# VideoTrace-R1: Long Video-based Retrieval-Augmented Generation via Reinforcement Learning

**Zongsheng Cao** [* 1]  **Anran Liu** [* † 2]  **Jun Xie** [3]  **Feng Chen** [3]  **Lang Chen** [2]  **Jing Li** [4]  **Zepeng Wang** [3]  **Zigan Wang** [† 5]

Long-video temporal reasoning remains a bottleneck for Large Video Language Models (LVLMs). Existing reinforcement-learning approaches reward only final-answer correctness, so they cannot distinguish answers reached through grounded reasoning from those reached through fabricated chronology; the intermediate temporal claims that constitute the reasoning are never verified. We trace this gap to a structural correspondence between two kinds of traces: a video has its own temporal trace, an ordered sequence of how events unfold, while a model's answer is built up through a reasoning trace, an ordered sequence of intermediate temporal claims. Correct reasoning requires the latter to mirror the former, claim by claim. We act on this correspondence with two contributions. We introduce Temporal Reasoning Traces (TRT), a structured index of a video's ordered event chains that exposes a small set of deterministic verification primitives, materializing the temporal trace as a programmatically queryable object. We then propose temporal-enhanced GRPO, a reinforcement-learning procedure whose reward decomposes into per-block components, each computed by a TRT primitive on a typed think block of the reasoning trace. Because the reward is fully symbolic, fabricated temporal claims are caught at the per-claim level rather than masked by a correct final answer. Across long-video reasoning benchmarks, our model achieves state-of-the-art performance, with the largest gains on out-of-domain reasoning tasks such as Video-Holmes, CG-Bench-Reasoning, and VRBench.

## 1. Introduction

Long-video understanding requires reasoning over hour-scale temporal contexts, where events unfold, influence each other, and form causal chains across extended dura-tions (Chen et al., 2023; Li et al., 2024a; Liu et al., 2024a; OpenGVLab, 2024). Recent Large Video Language Models (LVLMs) (Cheng et al., 2024; Chen et al., 2024b; Li et al., 2023b; Zhang et al., 2024c) have made multi-event reasoning increasingly tractable on short clips. As the temporal horizon grows, however, the bottleneck shifts: a single 30-minute recording can exceed 200K visual tokens (Bai et al., 2025b; Li et al., 2024a), the model's ability to keep events in their correct order degrades, and answers that look correct on the surface increasingly rest on fabricated chronology.

Two complementary lines of work have addressed long-video reasoning. Retrieval-augmented methods (Lewis et al., 2020; Luo et al., 2025; Cao et al., 2025a) narrow the input by selecting relevant clips before reasoning, with recent variants (Shen et al., 2025) organizing those clips by entity co-occurrence. Reasoning-enhanced LVLMs such as Video-R1 (Feng et al., 2025) and Time-R1 (Wang et al., 2025) instead train the model directly with reinforcement learning that rewards final-answer correctness. Both lines deliver real progress, but they share a structural deficit: neither materializes the video's temporal structure in a form that training can verify against, so the model's internal temporal logic remains a black box that the supervision signal never inspects.

This deficit produces two structural limitations of current systems, the second a consequence of the first. (C1) Temporal structure stays implicit. Whether a video is encoded as a flat token sequence or as an entity-co-occurrence graph, the order in which events unfold is not exposed in any form a downstream signal can query. A representation that records "person A" and "object B" as co-occurring across clips tells us little about whether "A picked up B after entering the room" or was already holding it before the door opened, yet that distinction is exactly what temporal questions turn on. (C2) Reasoning processes go unverified. Because temporal structure is unavailable as a structured object, no supervision signal can check whether a model's reasoning respects it. Outcome-only reinforcement learning rewards correct answers regardless of how they were reached, so a model can produce the right final answer through fabricated timestamps and orderings, and the reward has no mechanism to catch the fabrication. The result is a policy that looks

[*]Equal contribution   [†]Corresponding author. [1]Shanghai AI Laboratory. [2]Independent Researcher. [3]Lenovo Group. [4]Institute for Industrial Innovation and Finance, School of Economics and Management, Tsinghua University. [5]The University of Queensland. Correspondence to: <agiczsr@gmail.com, anniegogo1008@gmail.com, zigan.wang@uq.edu.au>.

*Proceedings of the 43rd International Conference on Machine Learning*, Seoul, South Korea. PMLR 306, 2026. Copyright 2026 by the author(s).

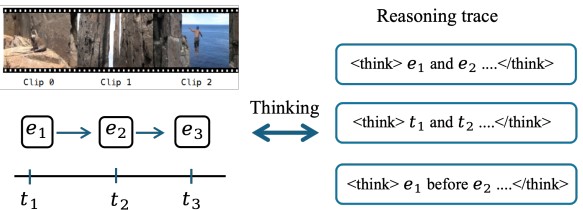

*Figure 1.* Temporal traces in video understanding. Events in videos form ordered temporal traces (left), which should be mirrored by the model's reasoning traces (right). VideoTrace-R1 explicitly models both, enabling verifiable temporal reasoning.

accurate but rests on incorrect intermediate claims.

This raises the natural question: *can we expose a video's temporal structure as a programmatically queryable object, and use it to verify a model's reasoning claim by claim?*

Our approach rests on a structural observation. A video has its own ordered sequence of events, a temporal trace (Figure 1, left). A model's answer is built up through an ordered sequence of intermediate claims, a reasoning trace (Figure 1, right). For temporal reasoning to be correct, the two traces must be in structural correspondence: each step of the reasoning trace should be answerable against the video's actual temporal trace. We materialize the former as Temporal Reasoning Traces (TRT), a queryable index of a video's ordered event chains that exposes a small set of deterministic verification primitives, and use these primitives to check the latter.

**VideoTrace-R1** instantiates this idea in two pieces. To address (C1), TRT explicitly indexes ordered event chains as queryable objects. Consider the question "What did the chef do between adding salt and serving?" Entity co-occurrence cannot recover the temporal arrangement of these actions, but TRT indexes chains such as $\langle\text{ADD\_SALT}\rangle \to \langle\text{STIR}\rangle \to \langle\text{SERVE}\rangle$ and exposes them through four boolean primitives that cover timestamp, entity, pairwise-order, and chain claims. To address (C2), we propose temporal-enhanced GRPO, a reinforcement-learning procedure in which the reasoning trace is structured into typed `<think>` blocks whose claims are routed to the matching primitive. The result is a fully symbolic, block-decomposed reward in which every signal is a deterministic query against TRT rather than the output of a learned scorer. Fabricated temporal claims therefore cannot quietly accrue reward; they are caught at the per-claim level and penalized.

We instantiate this design as a two-stage training procedure: supervised fine-tuning to internalize the typed-block format, followed by trace-grounded GRPO. Across long-video reasoning benchmarks including MLVU (Zhou et al., 2024), Video-MME (Fu et al., 2024), and Video-Holmes (Cheng et al., 2025), VideoTrace-R1 consistently improves over state-of-the-art 7B reasoning models, with the largest margins on out-of-domain reasoning tasks that demand multi-step temporal inference.

In summary, our contributions are threefold:

- We introduce temporal reasoning traces (TRT), a structured index that captures the ordered event chains of a video and exposes a small set of deterministic verification primitives, materializing the video's temporal trace as a programmatically queryable object.

- We propose temporal-enhanced GRPO, a reinforcement-learning procedure whose trace-grounded reward decomposes into per-block components, each computed by a TRT primitive on a typed think block of the model's reasoning trace. The reward is fully symbolic, so it directly penalizes temporally hallucinated reasoning that outcome-only signals cannot detect.

- Extensive experiments on long-video reasoning benchmarks show that VideoTrace-R1 achieves state-of-the-art performance, with the largest gains on out-of-domain temporal reasoning tasks such as Video-Holmes, CG-Bench-Reasoning, and VRBench.

## 2. Related Work

### 2.1. Video Understanding: From Images to Long Videos

With the development of artificial intelligence (Cao et al., 2025c;a; Huang et al., 2026), the evolution from static image understanding to temporal video analysis represents a fundamental paradigm shift in multimodal AI. Multimodal Large Language Models (MLLMs) (Chen et al., 2023; Zhu et al., 2023; Liu et al., 2024b;a; Tong et al., 2024) established the foundation by demonstrating powerful cross-modal reasoning between vision and language. This success naturally motivated the development of Large Video Language Models (LVLMs) (Ataallah et al., 2024a; Cheng et al., 2024; Li et al., 2023b; 2024b; Lin et al., 2023a; Luo et al., 2023), which incorporate temporal dynamics into multimodal understanding.

Contemporary LVLMs (Li et al., 2023b; Cao et al., 2025b) employ two architectural strategies for encoding video content. The first leverages alignment modules, typically adapted from BLIP-2's Q-Former (Li et al., 2023a), to establish sophisticated mappings between visual features and language representations (Li et al., 2023b; 2024b; Cheng et al., 2024). The second adopts direct concatenation of frame embeddings (Ataallah et al., 2024a; Lin et al., 2023a; Luo et al., 2023), prioritizing simplicity over learned alignment. Yet both approaches encounter a fundamental bottleneck: the context window constraint becomes prohibitive

when processing hour-long videos, where a 30-minute sequence can easily exceed 200K tokens (Shen et al., 2024; He et al., 2024). This limitation exposes a critical gap between current model capabilities and the demands of real-world long-video understanding.

## 2.2. Strategies for Handling Long-Context Videos

The research community has explored three distinct strategies to address the long-context challenge, each with inherent trade-offs.

**Compression-Based Approaches.** The most direct solution involves reducing video representation density through temporal subsampling (Li et al., 2023b; Ataallah et al., 2024a; Cheng et al., 2024; Zhang et al., 2024c; Li et al., 2024a), token-level pooling (Maaz et al., 2023; Li et al., 2023c; Song et al., 2023), or learned compression mechanisms (Shen et al., 2024; He et al., 2024). While computationally efficient, these methods inevitably discard fine-grained temporal details critical for understanding complex event sequences and causal relationships that span extended durations.

**Tool-Augmented Approaches.** Agent-based systems (Lin et al., 2023b; Wang et al., 2024c; Ma et al., 2024; Wang et al., 2024d; Fan et al., 2024; Zhang et al., 2024a) decompose video analysis into specialized subtasks, orchestrating multiple foundation models to handle different aspects of understanding. For instance, MM-VID (Lin et al., 2023b) employs GPT-4V (OpenAI, 2023) to generate textual video descriptions, while VideoAgent (Wang et al., 2024c) maintains a unified memory structure for tool coordination. Despite achieving strong empirical results, these systems incur substantial computational costs and critically depend on proprietary APIs, limiting their accessibility and raising concerns about data privacy and reproducibility.

**Retrieval-Augmented Approaches.** Retrieval-Augmented Generation (RAG) offers a third pathway by selectively retrieving relevant video segments rather than processing entire sequences. This paradigm decomposes the problem into knowledge indexing, query-driven retrieval, and context-aware generation, enabling models to focus computational resources on pertinent content. Our work shares the goal of focusing computational resources on pertinent temporal content, but rather than treating retrieval as the core mechanism, we use a structured index of the video's temporal trace to supervise reinforcement learning at the level of individual claims.

## 2.3. Graph-Based Video Understanding

Graph-structured representations have gained traction in the RAG paradigm as an alternative to flat, chunk-based retrieval (Gao et al., 2023; Chan et al., 2024). General-purpose knowledge graphs (Edge et al., 2024; Guo et al., 2024; Li et al., 2024c; Han et al., 2024) organize information via explicit entity-relation structures, enabling sophisticated retrieval beyond semantic matching. In video understanding, graph-based methods have proven effective for action recognition (Wang et al., 2021) and scene understanding (Hussein et al., 2019; Luo et al., 2025), where inter-object relationships are crucial.

However, applying graph-based RAG (Ren et al., 2025; Cao et al., 2026) to long videos introduces challenges. Early systems like Goldfish (Ataallah et al., 2024b) treated video clips as independent, missing cross-segment dependencies. Recent works (Ren et al., 2025; Luo et al., 2025; Cao et al., 2025a; Shen et al., 2025) have advanced by creating entity-centric graphs, with nodes representing clips and edges encoding co-occurrences, enabling retrieval based on inter-clip relationships.

While entity-centric graphs advance video RAG, they fail to capture temporal dynamics, treating causality and temporal order as secondary. In contrast, our framework introduces *ordered event paths* as first-class graph objects. Instead of linking clips by shared entities, we construct and index temporal-relational pathlets, directly queryable sequences like "person A enters BEFORE person B leaves." This path-centric approach enables structure-aware indexing, exposing events by their role in temporal chains, and addresses the key limitation of entity-centric video graphs by making temporal ordering explicit.

Our framework departs from prior long-video reasoning along two dimensions. First, outcome-supervised reasoning LVLMs such as Video-R1 (Feng et al., 2025), Time-R1 (Wang et al., 2025), and VideoChat-R1 (Li et al., 2025b) apply reinforcement learning that rewards only final-answer correctness, leaving the intermediate temporal claims of the reasoning trace unsupervised; a model that arrives at the right answer through fabricated chronology obtains the same reward as one that reasons faithfully. We instead route every intermediate temporal claim through deterministic queries against TRT, eliminating the neural reward step that itself introduces hallucination risk. Second, while the graph-based video representations discussed above record entity co-occurrence within a clip, they leave event order implicit; TRT lifts ordered event chains into a queryable index and exposes them through verification primitives that any temporal claim can be checked against. This shift in supervision granularity, from final answer to individual claim, is what drives the empirical gains we report in our experiments, particularly on out-of-domain reasoning where temporal hallucination is most prevalent. More broadly, our framework can be read as bridging two existing perspectives: the structured-graph view of video understanding and the process-supervision view of reasoning RL, with TRT

serving as the structural object that lets both operate deterministically.

# 3. Method

VideoTrace-R1 rests on a single structural premise about long-video reasoning. Long videos are not just collections of frames or entities; their content is shaped by the order in which events unfold, and any answer to a temporal question is implicitly a claim about that order. A video unfolds as an ordered sequence of events; a model's answer is built up through an ordered sequence of intermediate claims about what happened and when. We posit a simple correspondence between the two: *a model's reasoning trace is correct only if it mirrors the video's underlying temporal trace*. This trace-to-trace duality, illustrated in Figure 1, organizes our entire framework. To act on it we need both a representation of the video's temporal trace that is structured enough to query, and a learning signal that checks whether the model's reasoning trace lines up with that structure.

These two needs map onto our two contributions. Section 3.1 introduces temporal reasoning traces, a structured index that captures ordered event chains and exposes a small set of verification primitives. Section 3.2 introduces temporal-enhanced GRPO, a reinforcement-learning procedure whose reward is computed by routing every claim made along the model's reasoning trace through TRT's primitives, turning each step of inference into something that can be deterministically checked against the video's actual temporal structure. Figure 2 summarizes the full pipeline end-to-end.

## 3.1. Temporal Reasoning Traces

The first contribution is a representation that takes *ordered event chains* seriously. Conventional video features describe what is in a frame, while entity-centric video graphs describe *which* entities co-occur. Neither captures the basic fact that events have an order, and that this order is often the answer.

**Definition 3.1** (Temporal Reasoning Trace). A TRT index for a video $V$ is a tuple $\mathcal{T} = (\mathcal{E}, \mathcal{R}, \mathcal{I})$ where: $\mathcal{E}$ is an entity registry containing canonical entities $\{e_j\}$ with their descriptions and temporal spans; $\mathcal{R}$ is a trace repository storing temporal traces, each $\tau = (\pi, \mathcal{S}_\pi, [t_s, t_e], s)$ comprising an ordered event chain $\pi = \langle e_0 \xrightarrow{r_1} e_1 \cdots \xrightarrow{r_L} e_L \rangle$ ($L \leq 3$), supporting evidence $\mathcal{S}_\pi$, time interval $[t_s, t_e]$, and confidence score $s \in (0, 1]$; $\mathcal{I}$ is a temporal index enabling efficient time-based and entity-based queries.

Consider the question: "What did the chef do *between* adding salt and serving the dish?" Answering it requires not just knowing which entities appeared in which clip, but knowing their *order*. A trace like $\langle \text{ADD\_SALT} \rangle \xrightarrow{\text{THEN}}$ $\langle \text{STIR\_POT} \rangle \xrightarrow{\text{THEN}} \langle \text{SERVE} \rangle$ makes this ordering a first-class, queryable object. Two clips can share exactly the same entity set yet differ entirely in event order; lifting order itself into the representation is what lets us tell them apart.

**Dual-channel trace extraction.** We construct the TRT index through two synchronized channels using a frozen LVLM.

**(i) Entity channel.** For each video segment with frames $F_i$ and subtitles $C_i$, we extract entity mentions with temporal grounding:

$$\mathcal{M}_i = \{(e_j^i, d_j^i, [t_{s,j}^i, t_{e,j}^i])\} \leftarrow \text{LVLM}_{\text{ent}}(F_i, C_i), \quad (1)$$

where $e_j^i$ is the entity mention, $d_j^i$ its description, and $[t_{s,j}^i, t_{e,j}^i]$ its active time interval. Mentions are canonicalized via semantic similarity: if $\max_{e \in \mathcal{E}} \cos(d_j^i, d^e) \geq \epsilon$, we link to the existing canonical entity; otherwise, we instantiate a new entry.

**(ii) Trace channel.** Within a temporal window $\mathcal{W}_i$, we prompt the LVLM to extract ordered event relationships:

$$\mathcal{R}_i = \{(\pi_m^i, \mathcal{S}_{\pi_m^i}, [t_{s,m}^i, t_{e,m}^i])\} \leftarrow \text{LVLM}_{\text{trace}}(\mathcal{W}_i), \quad (2)$$

where each $\pi_m^i$ is an ordered event chain over canonical entities. Traces are deduplicated via signature matching and merged when structurally equivalent. Aggregating both channels across all segments yields the full TRT index $\mathcal{T} = (\mathcal{E}, \mathcal{R}, \mathcal{I})$, with each trace scored by extraction confidence and cross-segment consistency.

**TRT as a verification oracle.** TRT exposes four boolean primitives over the index: $\text{VERIFY\_TIME}(t)$ checks whether $t$ falls within any segment span recorded in $\mathcal{T}$; $\text{VERIFY\_ENTITY}(e)$ checks whether $e$ canonicalizes to some $e' \in \mathcal{E}$; $\text{VERIFY\_ORDER}(A, B)$ checks whether $A \prec B$ holds in some $\tau \in \mathcal{R}$; and $\text{VERIFY\_CHAIN}(\pi)$ checks whether $\pi$ matches a contiguous sub-chain of some $\tau \in \mathcal{R}$. These four queries cover the temporal claims a video question naturally gives rise to, and every answer is returned symbolically by consulting the index.

## 3.2. Temporal-Enhanced GRPO with Trace-Grounded Rewards

The second contribution operationalizes the trace-to-trace duality on the learning side. We adopt one design principle for the reward function: *every intermediate claim a model makes along its reasoning trace must be answerable by a deterministic query against the video-side index of §3.1*. Reinforcement learning that scores only final answers cannot enforce this principle, because the reward has no purchase on the body of the trace. Reward functions that score the

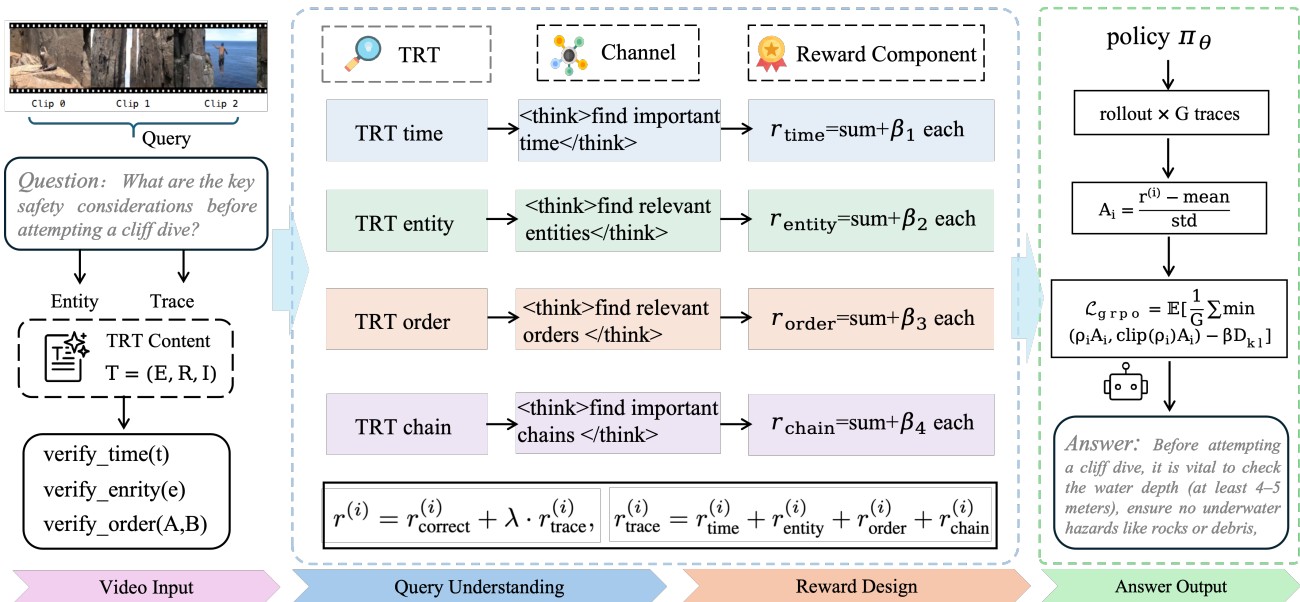

**Figure 2.** Overview of VideoTrace-R1. (Left) TRT is built offline by routing each video through a frozen LVLM with two extraction channels (entity and trace), producing an index $\mathcal{T} = (\mathcal{E}, \mathcal{R}, \mathcal{I})$ that exposes four boolean verification primitives. (Center) The model's reasoning trace is structured into four typed `<think>` blocks, each verified by the corresponding primitive from $\mathcal{T}$, yielding a block-decomposed reward $r_{\text{trace}}$ whose four components stand in one-to-one correspondence with the four primitives and the four think blocks. (Right) The composite reward $r = r_{\text{correct}} + \lambda \cdot r_{\text{trace}}$ enters standard GRPO to update the policy $\pi_\theta$. Every reward signal is a deterministic query against the index, with no learned scoring step.

trace through neural means inherit the very hallucination problem they are meant to detect. We therefore route every reward signal that touches the reasoning trace through TRT's verification primitives, producing a fully symbolic process reward.

**Reasoning trace as verification-typed thoughts.** We require the model to emit its reasoning as a structured trace of four `<think>` blocks, each typed by the TRT primitive it answers to:

- a `<think>` block carrying timestamp claims, verified by VERIFY_TIME;

- a `<think>` block carrying entity and state claims about what is present in the referenced interval, verified by VERIFY_ENTITY;

- a `<think>` block carrying pairwise ordering claims linking events across intervals, verified by VERIFY_ORDER;

- a `<think>` block carrying multi-step chain claims, verified by VERIFY_CHAIN.

The block scheme is therefore not a stylistic choice. Each block is a typed proposition language whose statements are exactly those TRT can answer. This is the first leg of a three-way correspondence at the core of our method: TRT

primitives, reasoning-trace blocks, and reward components stand in one-to-one alignment. The remaining two legs are closed in the paragraphs below.

**Stage 1: Supervised pretraining of the typed-block format.** Before reinforcement learning can verify block-specific claims, the model must reliably produce traces in the typed-block format. We perform supervised fine-tuning on a curated dataset $\mathcal{D}_{\text{SFT}}$ containing $(V, Q, T, Y)$ tuples with ground-truth reasoning traces:

$$\mathcal{L}_{\text{SFT}}(\theta) = -\mathbb{E}_{(V,Q,T,Y)} \Big[ \sum_{t=1}^{|[T;Y]|} \log p_\theta([T;Y]_t \mid V, Q, [T;Y]_{<t}) \Big]. \tag{3}$$

This stage installs the format. By itself it does not install grounded reasoning, which is the role of Stage 2.

**Stage 2: Trace-grounded GRPO.** We then apply Group Relative Policy Optimization (GRPO) with a reward that couples outcome supervision to trace-level verification:

$$r^{(i)} = r^{(i)}_{\text{correct}} + \lambda \cdot r^{(i)}_{\text{trace}}, \tag{4}$$

where $r^{(i)}_{\text{correct}} \in \{0, 1\}$ is the standard outcome reward indicating whether the predicted answer matches the ground truth, $r^{(i)}_{\text{trace}}$ scores the temporal grounding of the reasoning

trace (defined below), and $\lambda$ is a scalar coefficient balancing outcome correctness against process-level grounding.

**TRT-grounded trace reward.** At the heart of the reward is $r_{\text{trace}}$, which decomposes into four components in one-to-one correspondence with the typed blocks and TRT primitives:

$$r_{\text{trace}}^{(i)} = r_{\text{time}}^{(i)} + r_{\text{entity}}^{(i)} + r_{\text{order}}^{(i)} + r_{\text{chain}}^{(i)}, \qquad (5)$$

with each component computed by a single primitive call on every claim in the matching block of $T^{(i)}$: $r_{\text{time}}$ aggregates VERIFY_TIME over timestamps in the time block ($+0.1$ per grounded reference); $r_{\text{entity}}$ aggregates VERIFY_ENTITY over entity mentions in the entity block ($+0.1$ per verified entity); $r_{\text{order}}$ aggregates VERIFY_ORDER over each pairwise ordering asserted in the order block ($+0.2$ per correctly ordered pair); $r_{\text{chain}}$ aggregates VERIFY_CHAIN over multi-step chains in the chain block ($+0.3$ per matching chain). This closes the three-way correspondence: the reward is a typed sum whose components are both defined by, and verified against, the index from §3.1, with no learned scoring step in between.

**Per-block gradient flow.** A monolithic reward over the reasoning trace would conflate distinct error modes: a wrong timestamp, an unknown entity, a flipped pairwise ordering, and a fabricated multi-step chain are different mistakes calling for different fixes. Per-component rewards give per-block gradient flow, so the policy can learn to repair each kind of failure independently rather than averaging them away.

**GRPO optimization.** For each sample $(V, Q, Y)$, GRPO generates $G$ candidate traces and computes group-normalized advantages:

$$A_i = \frac{r^{(i)} - \text{mean}(\{r^{(j)}\}_{j=1}^G)}{\text{std}(\{r^{(j)}\}_{j=1}^G)}. \qquad (6)$$

The policy is updated by maximizing

$$\mathcal{J}_{\text{GRPO}}(\theta) = \mathbb{E}\left[ \frac{1}{G} \sum_{i=1}^G \Big( \min\big(\rho_i A_i, \text{clip}(\rho_i) A_i\big) - \beta D_{\text{KL}}^i \Big) \right], \qquad (7)$$

where $\rho_i = \pi_\theta(T^{(i)}|V,Q)/\pi_{\theta_{\text{old}}}(T^{(i)}|V,Q)$. Algorithm 1 summarizes the full procedure: TRT is built once per video offline, after which each iteration generates $G$ candidate traces, queries TRT to compute the per-component rewards, and updates the policy via GRPO.

**Closing the correspondence.** §3.1 makes the video's temporal trace *queryable*; this section makes the model's reasoning trace *verifiable*. Together they enforce the structural correspondence on which our notion of correctness depends:

---

**Algorithm 1** Temporal-Enhanced GRPO with TRT

**Require:** Video $V$, query $Q$, policy $\pi_\theta$
1: $\mathcal{T} = (\mathcal{E}, \mathcal{R}, \mathcal{I}) \leftarrow$ BuildTRT($V$) {offline, once per video; §3.1}
2: **for** each training iteration **do**
3:     Sample $G$ candidate traces $\{T^{(i)}\}_{i=1}^G \sim \pi_\theta(\cdot|V,Q)$
4:     **for** $i = 1, \ldots, G$ **do**
5:         Parse typed think blocks of $T^{(i)}$
6:         $r_{\text{time}}^{(i)} \leftarrow \sum$ VERIFY_TIME$(\cdot; \mathcal{T})$ on time block
7:         $r_{\text{entity}}^{(i)} \leftarrow \sum$ VERIFY_ENTITY$(\cdot; \mathcal{T})$ on entity block
8:         $r_{\text{order}}^{(i)} \leftarrow \sum$ VERIFY_ORDER$(\cdot; \mathcal{T})$ on order block
9:         $r_{\text{chain}}^{(i)} \leftarrow \sum$ VERIFY_CHAIN$(\cdot; \mathcal{T})$ on chain block
10:        $r^{(i)} \leftarrow r_{\text{correct}}^{(i)} + \lambda \cdot (r_{\text{time}}^{(i)} + r_{\text{entity}}^{(i)} + r_{\text{order}}^{(i)} + r_{\text{chain}}^{(i)})$
11:     **end for**
12:     Compute advantages $A_i = (r^{(i)} - \bar{r})/\sigma_r$
13:     Update $\theta$ via the GRPO objective $\mathcal{J}_{\text{GRPO}}$
14: **end for**

---

a reasoning trace earns its reward only when each of its temporal claims is grounded in TRT, so the policy is pushed toward traces that mirror, step by step, the order in which events actually unfold in the video.

**Remark on usage modes.** Although our primary instantiation trains a policy with trace-grounded GRPO, the framework decomposes into two reusable pieces that may be deployed independently. The trained policy is a stand-alone long-video reasoner. Separately, the TRT index of §3.1 is itself a queryable object that can supply structured temporal evidence to any other base LVLM, including as an external knowledge source in retrieval-augmented generation pipelines, without any parameter updates to the host model. The two uses share the same index but no parameters, and either or both may be deployed for a given task.

## 4. Experiments

### 4.1. Experimental Setup

**Datasets and Benchmarks.** Our evaluation spans both in-domain and out-of-domain settings to thoroughly probe VideoTrace-R1's temporal reasoning capabilities. For in-domain testing, we reserve 10% of each training source (ActivityNet (Caba Heilbron et al., 2015), LVBench (Wang et al., 2024b), ScaleLong (Ma et al., 2025), Star (Wu et al., 2024), and YouCook2 (Zhou et al., 2018)) as held-out evaluation splits; TutorialVQA (Colas et al., 2019) is excluded due to its limited size (76 samples). Out-of-domain generalization is measured on three challenging reasoning benchmarks unseen during training: Video-Holmes (Cheng et al., 2025), CG-Bench-Reasoning (Chen et al., 2024a), and VR-

*Table 1.* Comparison of model performance on video reasoning datasets in both in-domain and out-of-domain settings. The best results are marked in **bold** and the second best are underlined.

| Model | Out of Domain | | | In Domain | | | | |
|---|---|---|---|---|---|---|---|---|
| | **Video-Holmes** | **CG-Bench-Reasoning** | **VRBench** | **ActivityNet** | **Star** | **ScaleLong** | **YouCook2** | **LVBench** |
| *Open-source Vanilla Models* | | | | | | | | |
| InternVL-2.5-8B | 20.52% | 19.39% | 26.74% | 45.52% | 49.85% | 26.81% | 40.84% | 23.91% |
| InternVL-3-8B | 18.67% | 24.23% | 41.14% | 48.56% | 51.34% | 29.34% | 51.15% | 25.93% |
| Qwen2.5-VL-7B-Instruct | 34.02% | 27.10% | 63.42% | 70.96% | 69.25% | 40.06% | 63.74% | 33.33% |
| Qwen2.5-Omni-7B | 29.99% | 23.85% | 49.04% | 63.92% | 59.40% | 36.91% | 54.58% | 31.65% |
| *Open-source Reasoning Models* | | | | | | | | |
| Temporal-R1-7B | 33.81% | 25.27% | 60.92% | 70.88% | 70.15% | 39.75% | 63.74% | 32.66% |
| Open-R1-Video-7B | 21.83% | 16.46% | 50.15% | 55.76% | 44.48% | 31.86% | 50.76% | 26.94% |
| TW-GRPO-7B | 33.32% | 22.11% | 53.46% | 70.00% | 71.04% | 39.12% | 63.74% | 29.97% |
| Video-R1-7B | 38.54% | 27.81% | 69.25% | 76.00% | 67.76% | 47.32% | 65.65% | 34.68% |
| Time-R1-7B | 34.73% | 28.28% | 66.48% | 72.00% | 70.44% | 44.47% | 64.50% | 32.65% |
| VideoChat-R1-7B | 35.65% | 29.26% | 67.65% | 70.88% | **73.13%** | 40.69% | 69.08% | 32.99% |
| VideoChat-R1-Thinking-7B | 37.45% | 29.44% | 67.81% | 70.88% | 71.64% | 41.95% | 66.79% | 35.01% |
| GRPO-CARE-7B | 34.34% | 27.49% | 66.39% | 70.96% | 71.34% | 40.69% | 68.32% | 33.33% |
| VersaVid-R1-7B | 37.07% | 28.58% | 67.72% | 71.94% | 72.32% | 40.69% | 66.79% | 34.34% |
| VideoRFT-7B | 24.39% | 23.77% | 61.54% | 47.85% | 50.72% | 36.59% | 58.40% | 26.94% |
| VR-Thinker-7B | 25.37% | 19.54% | 53.43% | 55.36% | 63.58% | 32.18% | 51.91% | 30.98% |
| Video-RTS-7B | 29.56% | 18.09% | 27.71% | 63.60% | 65.07% | 30.28% | 65.27% | 20.88% |
| VideoTrace-R1-SFT-7B | 30.87% | 23.28% | 60.97% | 70.36% | 63.25% | 42.18% | 55.31% | 34.36% |
| VideoTrace-R1-7B | **43.51%** | **31.10%** | **76.23%** | **76.67%** | 66.87% | **48.32%** | 71.15% | **35.98%** |

Bench (Yu et al., 2025).

**Baseline Models.** We benchmark against two model families: (i) *vanilla LVLMs* (InternVL-2.5-8B (Chen et al., 2024c), InternVL-3-8B (Zhu et al., 2025), Qwen2.5-VL-7B-Instruct (Bai et al., 2025a), and Qwen2.5-Omni-7B (Xu et al., 2025)), which lack explicit reasoning supervision; and (ii) *reasoning-augmented LVLMs* (Temporal-R1-7B (Li et al., 2025a), Open-R1-Video-7B (Wang & Peng, 2025), TW-GRPO-7B (Dang et al., 2025), Video-R1-7B (Feng et al., 2025), Time-R1-7B (Wang et al., 2025), VideoChat-R1-7B and its Thinking variant (Li et al., 2025b), and GRPO-CARE-7B (Chen et al., 2025)), which incorporate various forms of reinforcement-based reasoning training.

**Training Details.** Starting from Qwen2.5-VL-7B-Instruct (Bai et al., 2025a), we first perform supervised fine-tuning on VideoTrace-R1-10K for one epoch (learning rate $1 \times 10^{-5}$, batch size 16). The model then undergoes GRPO training with KL coefficient $\beta = 0.04$, learning rate $5 \times 10^{-6}$, batch size 8, weight decay 0.01, and gradient clipping at norm 5. A unified prompt template governs both stages. For efficiency, videos are subsampled to at most 16 frames, each processed at $128 \times 28 \times 28$ resolution.

### 4.2. Training Mode Results

Table 1 presents our training-mode evaluation, where VideoTrace-R1 undergoes SFT followed by GRPO with trace-grounded process supervision. This setting directly tests whether verifiable temporal rewards translate into improved reasoning.

Across both in-domain and out-of-domain benchmarks, VideoTrace-R1-7B establishes a new state of the art among 7B-scale models (Table 1), claiming the top score on the majority of evaluated datasets and remaining competitive on the rest. The gains are most pronounced on out-of-domain benchmarks that demand complex temporal inference (Video-Holmes, CG-Bench-Reasoning, and VR-Bench), where our model surpasses the strongest reasoning-augmented baselines by sizable margins. This pattern indicates that trace-grounded supervision generalizes beyond training distributions and sharpens the model's ability to ground events, caption them faithfully, and chain them into coherent temporal narratives across long videos.

*Table 2.* **Performance comparison with LVLMs.** VideoTrace-R1 consistently improves all models on MLVU (Zhou et al., 2024), enhancing LongVU by 4.7% and Qwen2.5VL (7B) by 2.8%. Notably, VideoTrace-R1achieves 69.9% accuracy on Qwen2.5VL (3B), surpassing its 7B counterpart and improving the base model by 3.7%. VideoTrace-R1 outperforms base models across all video lengths on VideoMME (Fu et al., 2024) achieving improvement of 1.7% overall.

| Models | Size | MLVU | VideoMME | | LVB |
| --- | --- | --- | --- | --- | --- |
| | | | *w/o sub.* | *w/ sub.* | |
| *Proprietary LVLMs* | | | | | |
| Gemini 1.5 Pro (Google, 2024) | - | - | 75.0 | 81.3 | 64.0 |
| GPT-4o (OpenAI, 2024) | - | 64.6 | 71.9 | 77.2 | 66.7 |
| *Open-Source LVLMs* | | | | | |
| InternVL2.5 (Chen et al., 2024c) | 2B | 56.7 | 49.5 | 55.2 | 52.0 |
| InternVL2.5 + VideoTrace-R1 | 2B | 60.7 | 50.7 | 56.2 | 54.1 |
| Qwen2.5-VL (Bai et al., 2025b) | 3B | 66.2 | 61.4 | 67.6 | 54.2 |
| Qwen2.5-VL + VideoTrace-R1 | 3B | 69.9 | 62.7 | 69.0 | 57.8 |
| LongVU (Zhang et al., 2024b) | 7B | 65.4 | 55.2 | 60.9 | 50.2 |
| LongVU + VideoTrace-R1 | 7B | 70.1 | 56.9 | 63.5 | 52.2 |
| Qwen2-VL (Wang et al., 2024a) | 7B | 65.7 | 62.7 | 68.1 | 55.6 |
| Qwen2-VL + VideoTrace-R1 | 7B | 70.0 | 63.3 | 69.4 | 58.1 |
| LLaVA-Video (Zhang et al., 2024d) | 7B | 69.5 | 64.3 | 69.2 | 59.5 |
| LLaVA-Video + VideoTrace-R1 | 7B | 71.9 | 66.0 | 70.8 | 62.2 |
| Qwen2.5-VL (Bai et al., 2025b) | 7B | 68.8 | 65.1 | 71.1 | 56.0 |
| Qwen2.5-VL + VideoTrace-R1 | 7B | 71.6 | 68.5 | 73.6 | 59.6 |

Contrasting the two training stages reveals a striking pattern: GRPO with trace-grounded rewards delivers substantially stronger generalization than SFT alone. On every out-of-domain benchmark, the full model improves over the SFT-only checkpoint by a wide margin, with the gap widening as task complexity grows. Notably, VideoTrace-R1-SFT-7B often trails the Qwen2.5-VL-7B-Instruct baseline on out-of-domain reasoning tasks, suggesting that pure imitation of structured traces cannot, on its own, unlock robust cross-domain transfer. Yet SFT remains indispensable as a foundation: it endows the model with the structured trace format that GRPO can then refine through verifiable rewards. Without this scaffold, GRPO has no well-defined process to supervise. Together, these observations motivate our two-stage curriculum: SFT for format learning, followed by GRPO for generalizable temporal reasoning grounded.

### 4.3. TRT-Augmented Frozen LVLMs

The contribution of TRT (§3.1) is conceptually independent of the GRPO procedure (§3.2): TRT is a video-side index whose verification primitives can be queried by any consumer, not only our reward function. To probe whether TRT carries standalone value, we evaluate a complementary deployment in which TRT-extracted event chains are appended to the prompt of a frozen LVLM as additional context, with no parameter updates to the host model.

Table 2 reports results across six host backbones spanning 2B to 7B parameters on MLVU, VideoMME (with and without subtitles), and LongVideoBench. Two observations

stand out. First, TRT-augmented context yields consistent improvements over the unaugmented host on every backbone and every benchmark, indicating that TRT supplies inductive bias genuinely complementary to the host model rather than redundant with its own representations. Second, the smaller 3B Qwen2.5-VL host paired with TRT surpasses the larger 7B Qwen2.5-VL host without TRT on MLVU, suggesting that explicit temporal-structural information can substitute for a non-trivial amount of raw model capacity. Together with the training-mode results, these observations indicate that TRT is not merely a reward oracle for GRPO; its value persists outside the training loop, supporting our claim that TRT is a stand-alone abstraction rather than auxiliary infrastructure.

### 4.4. Ablation Study

*Table 3.* **Ablation study** of our VideoTrace pipeline on three benchmarks.

| Models | MLVU | VideoMME | LongVideoBench |
| --- | --- | --- | --- |
| VideoTrace w/o TRT | 69.7 | 72.3 | 62.7 |
| VideoTrace w/o GRPO (SFT only) | 71.2 | 73.5 | 65.0 |
| VideoTrace (full) | **72.5** | **74.7** | **66.1** |

**Ablation analysis.** Table 3 isolates the contribution of the two main components of VideoTrace-R1. Removing TRT (*w/o TRT*), so that the reward has no video-side index to query, produces the largest drop on all three benchmarks; this confirms that explicit ordered event chains are central to long-video temporal reasoning rather than an auxiliary feature. Skipping the GRPO stage (*w/o GRPO*) and using the SFT checkpoint alone also degrades performance, indicating that trace-grounded rewards provide supervision beyond what supervised imitation of the typed-block format captures. The combination of both is required: TRT supplies the structured target, and GRPO with block-decomposed rewards is what teaches the policy to align its reasoning trace to it.

## 5. Conclusion

We presented VideoTrace-R1, a framework for long-video temporal reasoning organized around the structural correspondence between a video's temporal trace and a model's reasoning trace. The framework materializes the video's temporal trace as a programmatically queryable object, and temporal-enhanced GRPO routes the reinforcement-learning reward through TRT, decomposing it into per-block components verified symbolically against the index. Across long-video reasoning benchmarks, VideoTrace-R1 outperforms strong reasoning-RL baselines, with the largest gains on out-of-domain tasks that require multi-step temporal inference.

## Impact Statement

This work advances video understanding by introducing trace-grounded process supervision for temporal reasoning in Large Video Language Models (LVLMs). By enabling models to provide verifiable, step-by-step temporal reasoning rather than opaque predictions, our approach contributes to more trustworthy and interpretable AI systems. The potential positive societal impacts include improved video content analysis for accessibility, education, and media understanding. However, as with all video understanding technologies, there are risks of misuse in surveillance or misinformation detection contexts. We encourage responsible deployment with appropriate human oversight, and note that our method's emphasis on grounded, verifiable reasoning may actually help mitigate hallucination-related harms common in current video-language models. We do not anticipate any direct negative ethical consequences from this research.

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
