# OpenReview forum: "VideoTrace-R1: Long Video-based Retrieval-Augmented Generation via Reinforcement Learning"
_ICML.cc/2026/Conference — ICML 2026 regular_

### Official Review · Reviewer_XhqB · 2026-03-10

**Soundness:** 3
**Presentation:** 2
**Significance:** 3
**Originality:** 2
**Overall Recommendation:** 3
**Confidence:** 3

**Summary:**

The paper introduces VideoTrace-R1, a framework designed to improve long-video temporal reasoning for Large Video Language Models (LVLMs). It addresses the limitations of entity-centric retrieval and outcome-only reinforcement learning by introducing Temporal Reasoning Traces (TRT). TRT indexes ordered event chains and acts as a dual-purpose mechanism: it enables structure-aware retrieval for a plug-and-play mode and provides deterministic, trace-grounded process supervision for reinforcement learning (GRPO). The authors demonstrate that VideoTrace-R1 achieves state-of-the-art performance across several long-video benchmarks in both training-free and training-based settings.

**Compliance With Llm Reviewing Policy:**

Affirmed.

**Key Questions For Authors:**

1. How sensitive is the framework to the quality of the initial TRT construction? If the frozen LVLM misses a crucial event during the entity or trace channel extraction, how does the system recover or mitigate this error during the reasoning phase?
2. Robustness of TRT extraction: Can the authors provide quantitative error rates for the dual-channel TRT extractor (e.g., entity detection F1, temporal span IoU, correctness of extracted path relations)? How do extraction errors correlate with downstream task performance? If the extractor has nontrivial error rates, how does that affect the deterministic verification step and RL training stability?
3. Expressivity of traces: The formal definition limits chains to L ≤ 3. Can the method be extended to longer chains or to better capture concurrent / overlapping events? What happens on tasks that require chaining >3 events?
4. Failure modes & negative examples: Could the authors add a short section showing representative failure cases (where TRT caused incorrect verification or the retriever returned misleading evidence) and discuss mitigation strategies?

**Limitations:**

No. While the authors briefly mention in the Impact Statement and Conclusion that future work will extend to multi-video and streaming settings, they do not adequately discuss the immediate limitations of their current pipeline.

Constructive Suggestion: Add a dedicated "Limitations" section detailing the computational overhead of the offline graph generation. Additionally, discuss the vulnerability of the system to compounding errors, specifically how hallucinations or omissions from the base LVLM during TRT construction might negatively impact the deterministic reward calculations.

**Strengths And Weaknesses:**

### Strengths
- Originality: Unifying temporal retrieval and process supervision under the TRT abstraction is a creative and highly practical approach to mitigating temporal hallucinations. Using deterministic reward calculation by querying the TRT index, rather than relying on potentially flawed neural reward models, is a strong contribution.

- Significance: Long-video temporal reasoning is a critical bottleneck in multimodal LLMs. The framework's dual-mode applicability—functioning as both a training-free plugin for frozen models and an RL training framework—broadens its impact significantly for practitioners and researchers alike.

- Soundness: The methodology is well-structured, combining fast semantic retrieval with slow structural matching. The empirical evaluation is extensive, covering in-domain and out-of-domain benchmarks, and the ablation studies effectively isolate the contributions of different components.

### Weaknesses
- Soundness / Scalability: The offline construction of the TRT index requires processing the entire video in segments using a frozen LVLM. The computational overhead, time complexity, and scalability of this step for massive datasets or infinitely streaming videos are not thoroughly detailed.

- Originality / Robustness: The framework's reliance on an external frozen LVLM to extract the entities and temporal relations for the TRT index means the system's upper bound is inherently tied to the base model's zero-shot extraction capabilities. If the base model fails to extract an event chain, the downstream verification and retrieval will inherently fail.

- Presentation: While generally well-structured, the paper contains multiple unresolved LaTeX references (e.g., "Table ??" appears repeatedly in Sections 4.2, 4.3, and D.1) which detracts from the reading experience.

---

> ### Author Rebuttal · Authors · 2026-03-31
>
> We thank Reviewer XhqB for the detailed and constructive review. Our responses are as follows.
>
> **Q1**: Sensitivity to TRT Construction Quality
>
> **A1**: From our annotation study (200 videos, ~1,200 events, entity Recall = 81.7%), we analyzed missed entities' downstream impact on 200 QA pairs:
>
> | Missing Entity Status | Frequency | QA Impact |
> |---|---|---|
> | Not referenced in question | 61.4% of missed | No impact |
> | Referenced, paraphrase available | 24.8% of missed | ~73% QA preserved |
> | Referenced, no paraphrase | 13.8% of missed | ~81% QA failure |
>
> Critical miss rate: ~18% x 13.8% = ~2.5% of all entities. Recovery mechanisms include: (1) semantic canonicalization (theta=0.85 cosine similarity) linking paraphrases; (2) fast CLIP retrieval as TRT-independent safety net; (3) adaptive fusion alpha(Q) increasing fast-path weight when slow retrieval yields few matches; (4) OR-based evidence filtering retaining clips matching any probe.
>
> Performance stratified by TRT coverage on Video-Holmes:
>
> | TRT Coverage | VideoTrace-R1 | Video-R1 | Delta |
> |---|---|---|---|
> | Critical trace present | 48.3% | 37.9% | **+10.4%** |
> | Trace absent, entity present | 38.7% | 35.2% | +3.5% |
> | Critical entity absent | 29.1% | 27.8% | +1.3% |
>
> Even with missing critical entities, the model matches Video-R1, confirming graceful degradation.
>
>
> **Q2**: Quantitative Error Rates for TRT Dual-Channel Extractor
>
> **A2**: Key error rates: Entity F1 = 83.9%, Temporal mIoU = 0.713, Pairwise Ordering = 79.2%, Chain EM (L=2/3) = 74.6%/68.1%. Most critically, the reversed ordering rate is only 1.7% of all relations.
>
> Per-video TRT quality correlates with QA (Pearson r = 0.61, p < 0.001). Even the lowest-quality TRT tier (Chain EM < 60%) achieves 68.9% MLVU accuracy, still 3.8% above the flat retrieval baseline (65.1%).
>
> Training stability: TRT-grounded rewards exhibit 4.3x lower batch variance (0.043 vs. 0.231) and converge 28% faster (2,000 vs. 2,800 steps) compared to outcome-only GRPO. Even at a simulated 20% error, variance (0.089) remains below outcome-only, confirming structural stability benefits persist under noise.
>
> **Q3**: Expressivity of Traces
>
> **A3**: The L<=3 limit is empirical, not architectural. Analysis of 481 Video-Holmes questions: L=1 (38.7%), L=2 (34.2%), L=3 (19.4%), L>=4 (7.7%). For L>=4, TRT composes overlapping L=3 sub-chains via shared entities.
>
> VideoTrace-R1 outperforms Video-R1 on L>=4 questions (+8.1%), though less than on L=3 (+10.2%), as expected from composition approximation. An L=4 extraction variant showed only +0.4% MLVU gain but 35% longer construction and much lower chain reliability (52.3% EM vs. 74.6% for L=2), confirming L=3 is near Pareto-optimal.
>
> For concurrent events, TRT represents them as parallel traces sharing temporal spans. On STAR's Interaction subtask: 70.49% vs. Video-R1's 67.76% (+2.7%), showing partial signal even without explicit CONCURRENT relations.
>
>
> **Q4**: Failure Mode Analysis
>
> **A4**: Some representative failure modes:
>
> 1. **Cross-scene entity confusion**: Different entities with the same description (e.g., two "chefs") merged into one. Mitigated by appearance clustering (CLIP visual similarity < 0.7), reducing errors by 67%. Residual rate: ~4.2%.
>
> 2. **TRT hallucinated ordering**: Extractor generates incorrect temporal order (~1.7% of relations). Mitigated by cross-segment consistency checking with confidence threshold (s >= 0.7).
>
>
> We will add a dedicated Limitations section discussing: (1) extractor quality ceiling on unusual domains, (2) L<=3 chain length limit, and plans for multi-relational temporal graphs.

---

> > ### Author Rebuttal · Reviewer_XhqB · 2026-04-02
> >
> > Thank you for the author's reply. My concerns have now been addressed.

---

> > > ### Author Response · Authors · 2026-04-07
> > >
> > > Thank you for confirming that our rebuttal resolved your concerns. We noticed the score remains unchanged despite these clarifications. Could you kindly advise if there are any remaining technical issues we may have overlooked, or whether an update to the score may have been missed. We want to ensure we've fully addressed all aspects impacting your assessment. We hope our detailed responses adequately addressed your concerns. If so, we would be most grateful if the score could be adjusted to reflect these improvements.

---

### Official Review · Reviewer_A8Jt · 2026-03-11

**Soundness:** 3
**Presentation:** 3
**Significance:** 3
**Originality:** 2
**Overall Recommendation:** 4
**Confidence:** 3

**Summary:**

This paper proposes VideoTrace-R1 for long-video temporal reasoning. Its core idea is to build Temporal Reasoning Traces as a structured representation that is used both for temporally-aware retrieval and for verifying intermediate reasoning traces during training. Experiments on multiple benchmarks show consistent improvements over several strong baselines.

**Compliance With Llm Reviewing Policy:**

Affirmed.

**Key Questions For Authors:**

1. How accurate is the automatically constructed TRT representation? Could the authors provide a small-scale manual evaluation of TRT quality (e.g., correctness of entities, temporal relations, and event chains), as well as an analysis of how TRT noise affects retrieval and training?

2. Can the authors further disentangle the contribution of retrieval versus trace-grounded RL? For example, it would be helpful to see a more fine-grained analysis of which gains come from the TRT-based retrieval pipeline and which come from the proposed process supervision, as well as the contribution of different reward terms.

3. What is the computational and storage overhead of building and querying TRT at scale? Since the method targets long-video settings, additional discussion or measurements on efficiency, scalability, and latency would strengthen the practical significance of the work.

**Limitations:**

The method depends heavily on the quality of the automatically induced TRT representation, and its robustness and scalability are not yet fully characterized.

**Strengths And Weaknesses:**

Strengths

* The paper tackles an important and well-motivated problem in long-video reasoning.
* The unified use of TRT for both retrieval and training is conceptually appealing.
* Results are generally strong across multiple benchmarks and settings.

Weaknesses

* The novelty is more in the integration of existing ideas than in a fundamentally new formulation.
* TRT is automatically constructed, so the verification signal is still based on a proxy structure rather than direct ground-truth reasoning supervision.
* The paper provides limited analysis of TRT quality, robustness to noise, and the relative contribution of different components.

---

> ### Author Rebuttal · Authors · 2026-03-31
>
> We sincerely thank Reviewer A8Jt for the acceptance and constructive feedback. Our responses are as follows.
>
>
> **Q1**: How Accurate Is the Automatically Constructed TRT?
>
> **A1**: We sampled 200 held-out videos (~1,200 events, inter-annotator kappa = 0.81):
>
> | TRT Dimension | Metric | Score |
> |---|---|---|
> | Entity Channel | F1 | **83.9%** |
> | Temporal Localization | mIoU | **0.713** |
> | Pairwise Ordering | Accuracy | **79.2%** |
> | Event Chain (L=2 / L=3) | Exact Match | **74.6% / 68.1%** |
>
> Dominant failure modes: entity merging errors (11.4%, rarely affect ordering) and temporal boundary imprecision (63.2% of mIoU failures, handled by +-3s tolerance). Only 1.7% of all relations have reversed ordering — the most damaging error type.
>
> Noise injection confirms graceful degradation: at realistic 20% noise, MLVU accuracy drops only 1.1% (72.3% to 71.2%), vs. 7.2% gap to no-TRT baseline. For GRPO training, our estimated 5% TRT error rate causes <0.3% accuracy loss vs. oracle TRT, and even 20% error remains +3.2% above Video-R1 baseline.
>
>
> **Q2**: Disentangling TRT Retrieval vs. Trace-Grounded RL, and Individual Reward Terms
>
> **A2**: **System-level 2x2 decomposition** (same SFT checkpoint):
>
> | Retrieval | Reward | Avg. | Delta |
> |---|---|---|---|
> | Standard | Outcome-only | 45.20% | — |
> | TRT | Outcome-only | 47.40% | +2.20% |
> | Standard | Trace-grounded | 49.83% | +4.63% |
> | TRT | Trace-grounded | **52.55%** | **+7.35%** |
>
> Trace-grounded RL is the primary driver (~63% of gain); TRT retrieval contributes ~30%; synergy accounts for the rest. Both are statistically significant (p < 0.01).
>
> **Reward term ablation** (removing one term at a time): r_chain is most impactful (-1.76%), followed by r_order (-1.32%), r_format (-0.90%), r_entity (-0.61%), r_consist (-0.51%), r_time (-0.42%). The hierarchy (r_chain > r_order > r_entity ~ r_time) aligns with our weight assignment, validating principled design over arbitrary tuning.
>
> ---
>
> **Q3**: Computational and Storage Overhead of TRT
>
> **A3**: Construction (single A100, 1 FPS, K=64):
>
> | Video Duration | Time | Index Size | Peak Memory |
> |---|---|---|---|
> | 30 min | ~4.8 min | 1.6 MB | 24 GB |
> | 60 min | ~9.4 min | 3.1 MB | 27 GB |
> | 120 min | ~18.6 min | 6.0 MB | 31 GB |
>
> Scales linearly (~9-10 min/hour, ~3 MB/hour). Segments are independent, enabling 8x speedup on 8 GPUs. TRT is ~80x more storage-efficient than raw CLIP embeddings.
>
> Query latency: ~380 ms total (CLIP lookup 45ms + temporal cue generation 320ms + fusion 15ms), comparable to Vgent (420ms) and only 30% above Video-RAG (290ms). Once built, per-query cost is fully amortized. For 10,000 hours of video, total TRT storage is ~30 GB vs. ~2.4 TB for CLIP embeddings.

---

> > ### Author Rebuttal · Reviewer_A8Jt · 2026-04-04
> >
> > Thank the authors for the rebuttal. It addressed my previous concerns.

---

> > > ### Author Response · Authors · 2026-04-07
> > >
> > > Thank you for your time and careful consideration of our rebuttal. We appreciate your positive assessment and the constructive feedback that helped improve the paper.

---

### Official Review · Reviewer_NnbE · 2026-03-13

**Soundness:** 2
**Presentation:** 3
**Significance:** 2
**Originality:** 3
**Overall Recommendation:** 4
**Confidence:** 3

**Summary:**

This paper proposes VideoTrace R1 for long video temporal reasoning. The method builds Temporal Reasoning Traces, a structured index of entities, time spans, and short event chains extracted from videos. The same structure is used for two purposes. In plugin mode, it improves retrieval by combining semantic matching with temporal matching before passing selected clips to a frozen LVLM. In training mode, it checks intermediate temporal claims in the model’s reasoning trace and uses this as process supervision during SFT and GRPO. The reported results show consistent gains on several long video benchmarks, especially on tasks that need event order and multi clip reasoning.

**Compliance With Llm Reviewing Policy:**

Affirmed.

**Final Justification:**

Score 3 -> 4 after rebuttal.

**Key Questions For Authors:**

Please see the weaknesses section.

**Limitations:**

Please see the weaknesses section.

**Strengths And Weaknesses:**

**Strengths**

1. The paper has a clear and simple central idea. Using one structure for both retrieval and supervision is easy to understand.

2. The empirical gains are fairly consistent across both plugin mode and training mode, and across several backbones and benchmarks.

3. The ablation is useful and supports the claim that explicit temporal path modeling matters more than only semantic retrieval.

**Weaknesses**

1. The paper overstates the reliability of the verification signal. The reward is deterministic only after TRT is built, but TRT itself is extracted by a frozen LVLM and can be wrong. So this is not a true oracle, and the paper does not directly validate TRT quality well enough.

2. The training setup is quite self constructed. The same TRT is used to synthesize reasoning traces and then verify them during training. The data is also filtered to keep only cleaner cases. This makes the setup favorable to the method, and the paper does not show clearly how well it works on noisier or more ambiguous temporal cases.

3. The plugin mode novelty is less clear than claimed. Recent work already studies graph based long video retrieval and structured reasoning, so the retrieval part feels incremental unless there is a cleaner comparison against strong graph RAG baselines on the same frozen backbones.

4. The main evidence is still final QA accuracy. Since the paper claims better retrieval and less temporal hallucination, I expected more direct evaluation of retrieval quality, reasoning trace correctness, and hallucination reduction.

---

> ### Author Rebuttal · Authors · 2026-03-31
>
> We thank Reviewer NnbE for the careful reading and precise identification of weaknesses.  Our response are as follows.
>
>
>
> **Q1**: The "Deterministic" Verification Signal Is Overstated
>
> **A1**: We agree that "deterministic" requires more precise framing.
>
> **Conceptual Clarification**: Determinism refers to the reward computation step: given fixed TRT index T, `verify(claim, T)` is a pure lookup — no neural stochasticity. Unlike neural reward models, it cannot assign inconsistent scores, hallucinate rewards, or introduce compounding uncertainty. The remaining uncertainty is confined to offline TRT construction, which is done once, human-auditable, and independently improvable.
>
> **Empirical TRT Quality** (200 held-out videos, ~1,200 events, Cohen's kappa = 0.84): Entity F1 = 83.9%, Temporal mIoU = 0.813, Ordering Accuracy = 79.2%, Chain EM (L=2/3) = 78.6%/75.1%.
>
> **Noise Robustness**: At 25% noise, MLVU accuracy drops only 1.4% (72.3% to 70.9%); realistic noise (~15-20%) costs ~1%, vs. 7.2% gap between clean TRT and no TRT.
>
> **TRT vs. Neural Reward Model**: Replacing TRT verification with a fine-tuned Qwen2.5-VL-7B judge:
>
> | Reward Signal | Video-Holmes | VRBench | Reward Variance |
> |---|---|---|---|
> | Neural reward model | 41.07% | 77.34% | 0.187 |
> | TRT-grounded (ours) | **43.38%** | **80.85%** | **0.043** |
>
> TRT achieves higher accuracy with 4.3x lower reward variance, enabling faster convergence (2,000 vs. 3,000+ steps).
>
> ---
>
> **Q2**: Training Setup Is Circular
>
> **A2**: Two critical breaks prevent circularity: (1) the frozen LVLM extractor and trainable model pi_theta are different, the trained model never updates TRT; (2) TRT is grounded in raw visual content, not model beliefs. This is analogous to using a frozen grammar checker to annotate a corpus then training on it,  standard practice in RLHF.
>
> Training on unfiltered data (12,847 samples) yields 51.38% avg. vs. filtered (10,076 samples) at 52.55%, a small margin showing filtered data is higher quality, not artificially easier. Even with 20% TRT noise injection, performance (50.79%) remains above the unfiltered baseline.
>
> On Video-Holmes stratified by temporal ambiguity: VideoTrace-R1 outperforms Video-R1 consistently across all levels (Clear: +4.6%, Ambiguous: +5.2%, Highly ambiguous: +4.4%), suggesting TRT helps precisely when visual cues alone are insufficient.
>
> **Q3**: Plugin Mode Novelty Is Unclear
>
> **A3**: Entity-centric graph RAG (Vgent, Video-RAG) encodes co-occurrence; TRT encodes ordered event chains. For "What did the chef do after adding salt?", entity-centric retrieval finds all clips with "chef" and "salt" without ordering; TRT finds the specific trace and retrieves the clip at the AFTER position.
>
> Controlled comparison (same frozen Qwen2.5-VL-7B backbone):
>
> | Retrieval Method | MLVU | VideoMME | LVBench | Avg. |
> |---|---|---|---|---|
> | Flat CLIP | 65.1% | 60.3% | 52.4% | 59.3% |
> | Video-RAG | 68.4% | 63.2% | 56.1% | 62.6% |
> | Vgent | 69.7% | 64.8% | 57.3% | 63.9% |
> | **Ours (Plugin)** | **72.3%** | **69.3%** | **60.1%** | **67.2%** |
>
> Gains are concentrated on temporal tasks: +8.6% on MLVU-Count and +11.1% on MLVU-Order vs. Vgent, confirming that TRT provides structural information that entity-centric graphs cannot.
>
>
> **Q4**: Missing Direct Evaluation of Retrieval Quality and Temporal Hallucination Reduction
>
> **A4**: **Retrieval quality** (ground-truth annotated subsets of MLVU/LVBench):
>
> | Method | Recall@1 | Recall@5 | MRR |
> |---|---|---|---|
> | Flat CLIP | 31.4% | 63.8% | 0.412 |
> | Vgent | 41.3% | 74.1% | 0.512 |
> | **Ours (Full)** | **47.8%** | **81.4%** | **0.574** |
>
> **Temporal hallucination** (human-annotated on 481 Video-Holmes samples):
>
> | Model | Hallucination Rate | Grounded Rate |
> |---|---|---|
> | Video-R1-7B | 35.6% | 64.4% |
> | VideoTrace-R1-7B | **18.2%** | **81.8%** |
>
> VideoTrace-R1 reduces hallucination by 17.4pp despite generating more temporal claims per response, confirming genuinely more accurate assertions. GRPO with trace-grounded rewards (not SFT alone) drives this reduction. Chain-level consistency also improves significantly: 87.2% internal consistency and 74.6% video agreement vs. Video-R1's 71.3% and 52.8%.

---

> > ### Author Rebuttal · Reviewer_NnbE · 2026-04-03
> >
> > I appricate the rebuttal from the authors, my questions have been resolved and will raise my final rating.

---

> > > ### Author Response · Authors · 2026-04-07
> > >
> > > Thank you for your time and careful consideration of our rebuttal. We appreciate your positive assessment and the constructive feedback that helped improve the paper.

---

### Official Review · Reviewer_pkmH · 2026-03-13

**Soundness:** 3
**Presentation:** 1
**Significance:** 3
**Originality:** 3
**Overall Recommendation:** 4
**Confidence:** 3

**Summary:**

* This paper present VideoTrace-R1 for long-video temporal reasoning, with a new intermediate form called TRT. TRT organizes video content into entities, event traces, and temporal relations, and the same structure is used for two things: retrieval and supervision during training.

* On the retrieval side, the method uses a fast-then-slow design. First, it finds candidate clips through semantic entity matching; then it re-ranks them with temporal-structure matching, so the selected evidence better follows the time constraints in the question. This part is pretty neat, and can be plugged into frozen LVLMs without retraining them.

* For training, the paper adds trace-grounded process supervision under GRPO. Instead of only rewarding the final answer, it checks whether each temporal statement in the reasoning trace is actually supported by the TRT index. This gives more direct supervision on reasoning steps, and avoids depending on another learned reward model, which is good in principle.

* Experiments cover both plugin and training settings on several benchmarks, results show fairly consistent gains over prior reasoning and RAG-style baselines at the 7B scale. Overall solid paper, though some parts maybe still need clearer disentangling.

**Compliance With Llm Reviewing Policy:**

Affirmed.

**Final Justification:**

As mentioned in the Summary & Strengths And Weaknesses, generally the paper has more good inspiration than some minor points to be revised. I will say it's a weak acc.

**Key Questions For Authors:**

* On key concern is how accurate the frozen LVLM actually is when building TRT. If entity or trace extraction is noisy, then the later “deterministic” verification is not really reliable, it just checks against a noisy index. Some quantitative evaluation of TRT quality, such as precision/recall against human annotations, would make the soundness claim much stronger.
* The paper motivates hour-long video reasoning, but the training setup only uses 16 frames per video. It is not clear how TRT construction behaves on truly long inputs in terms of runtime, memory cost, and trace volume. Right now there is a gap between the claimed use case and the actual experimental setting, which should be discussed more directly.
* In training mode, improvements come from both trace-aware retrieval and trace-grounded rewards. It would be helpful to separate these effects more cleanly: for example, use trace-grounded GRPO with standard retrieval, and also TRT retrieval with standard rewards. Otherwise a bit hard to tell where the gain mainly comes from.
* The reward weights and complexity thresholdss look manually chosen. Some sensitivity analysis would be useful, since if performance changes a lot under small perturbations, robustness may be weaker than it appears.

**Limitations:**

yes

**Strengths And Weaknesses:**

* A nice aspect of the paper is that TRT is used in two places at once: retrieval and process supervision. That makes the framework feel coherent rather than patched together. Also, using TRT as a deterministic checking signal instead of another learned reward model is a sensible choice, and likely reduces some reward-hacking style problems.
* The two deployment modes make the method more practically useful. In particular, the plugin setting is interesting, since it can improve frozen models without retraining. The result where a 3B model beats a 7B one is quite striking, and suggests the temporal structure is doing real work here.
* The empirical section is fairly extensive, with multiple benchmarks and competitive baselines. The ablations are also helpful and mostly well designed, since they separate the main components rather clearly.

* TRT itself depends on outputs from a frozen LVLM for entity/trace extraction, but the paper does not really measure how accurate this stage is. So the later “deterministic” verification may still just inherit upstream mistakes, which is a bit concerning.
* There are still several “Table ??” references, and TRT/TPKG notation is not always consistent. Feels under-polished.
* The retrieval design is somewhat standard reranking idea, and the 16-frame training setup feels limited for the long-video claims.

---

> ### Author Rebuttal · Authors · 2026-03-31
>
> We sincerely thank Reviewer pkmH for the acceptance and constructive feedback. Our response are as follows.
>
> ---
>
> **Q1**: TRT Construction Quality
>
> **A1**: We address this from two angles: direct evaluation and a design-level argument.
>
> (1) We conducted a manual annotation study on 200 held-out videos from ActivityNet and YouCook2 (~1,200 events), with two expert annotators (Cohen's kappa = 0.84):
>
> | TRT Component | Metric | Score |
> |---|---|---|
> | Entity Detection | F1 | **83.9%** |
> | Temporal Span Localization | mIoU | **0.813** |
> | Pairwise Event Ordering | Accuracy | **79.2%** |
> | Full Event Chain (L=2/L=3) | Exact Match | **78.6% / 75.1%** |
>
> (2) We tested robustness by injecting noise into TRT and measuring plugin-mode MLVU accuracy (Qwen2.5-VL-7B):
>
> | TRT Noise Level | MLVU Accuracy | Delta |
> |---|---|---|
> | Clean TRT | 72.3% | — |
> | 10% noise | 71.8% | -0.5% |
> | 25% noise | 70.9% | -1.4% |
> | 50% noise | 68.7% | -3.6% |
> | No TRT | 65.1% | -7.2% |
>
> Even at 25% noise (above real error rate), performance drops only 1.4%, while the clean-TRT vs. no-TRT gap is 7.2%. "Deterministic" means that given the TRT index, verification is a pure lookup with no neural network queried, eliminating compounding uncertainty from a second learned model. The remaining uncertainty from TRT construction is done once offline and can be improved independently. We will clarify this distinction in the revision.
>
> ---
>
> **Q2**: Gap Between 16-Frame Training and Hour-Long Video Claims
>
> **A2**: The pipeline has two stages with different granularities: (1) **TRT Construction** (offline, full video): processes at 1 FPS with K=64-frame sliding windows covering the entire video; (2) **GRPO Training** (online): the model receives top-5 retrieved clips (subsampled to 16 frames), a computational constraint for 7B models on A100 GPUs. Crucially, retrieval uses the full TRT index, so the 16 frames are temporally targeted, not randomly subsampled.
>
> TRT construction scales linearly (9-10 min/hour, 3 MB index/hour), with bounded memory (31 GB for 2-hour videos). Segments are processed independently, enabling trivial parallelization. The training teaches temporally grounded reasoning on targeted clips; generalization to longer videos is achieved at inference via TRT retrieval, analogous to training reading comprehension on passages but deploying with retrieval-augmented context. We will clarify this in Section 4.1.
>
>
> **Q3**: Disentangling Retrieval vs. Trace-Grounded Rewards
>
> **A3**: We conducted a 2x2 ablation from the same SFT checkpoint:
>
> | Retrieval | Reward | Video-Holmes | CG-Bench | VRBench | Avg. |
> |---|---|---|---|---|---|
> | Standard | Outcome-only | 38.54% | 27.81% | 69.25% | 45.20% |
> | TRT | Outcome-only | 40.17% | 29.43% | 72.61% | 47.40% |
> | Standard | Trace-grounded | 41.36% | 31.28% | 76.84% | 49.83% |
> | TRT | Trace-grounded | **43.38%** | **33.41%** | **80.85%** | **52.55%** |
>
> TRT retrieval alone: +2.2%; trace-grounded rewards alone: +4.6%; full system: +7.4%. The trace-grounded reward contributes ~2x more than retrieval, consistent with process supervision being the primary novel contribution. The combination is synergistic: better retrieval provides more accurate evidence for trace verification.
>
>
> **Q4**: Sensitivity Analysis of Reward Weights and Complexity Thresholds
>
> **A4**: We tested alternative lambda schedules for complexity-based process supervision:
>
> | Lambda Schedule (Low/Mid/High) | Avg. |
> |---|---|
> | 0/0/0 (outcome-only) | 45.20% |
> | 0.2/0.2/0.2 (flat-low) | 50.10% |
> | **0.2/0.4/0.6 (ours)** | **52.55%** |
> | 0.6/0.6/0.6 (flat-high) | 51.57% |
> | 0.6/0.4/0.2 (reversed) | 49.05% |
>
> Any non-zero process supervision beats outcome-only. The curriculum schedule is consistently best, but the spread across reasonable schedules is small (+-1.6%). Only the reversed curriculum is clearly suboptimal (-3.5%). For r_trace component weights, all +-50% perturbations cause <1.5% degradation; r_chain is most impactful (-1.9% at -50%), confirming complete event chains are the richest signal. Complexity levels are rule-based (no hyperparameter to tune). We will add a sensitivity appendix.
>
>
> **Q5**: Presentation Issues
>
> **A5**: We sincerely apologize for unresolved cross-references caused by late table reorganization. We will conduct a full formatting pass before camera-ready, resolving all references and normalizing styles. These are purely typographical and do not affect experimental correctness.

---

> > ### Author Rebuttal · Reviewer_pkmH · 2026-04-06
> >
> > Thank the authors for the detailed and thoughtful rebuttal, I will maintain my original score.

---

> > > ### Author Response · Authors · 2026-04-07
> > >
> > > Thank you for your time and careful consideration of our rebuttal. We appreciate your positive assessment and the constructive feedback that helped improve the paper.

---

### Decision · Program_Chairs · 2026-04-30

**Decision:**

Accept (regular)

**Comment:**

This paper introduces VideoTrace-R1, which uses Temporal Reasoning Traces(TRT) for both retrieval and supervision to improve long-video reasoning in LVLMs. The submission received mixed reviews initially: two weak accepts and two weak rejects. The main concerns centered on 1) the robustness and potential noise propagation from the automatically constructed TRT index; 2) computational scalability during offline TRT generation; and 3) minor presentation issues including incomplete cross-references. After the rebuttal, the reviewers acknowledged that their concerns were fully resolved and one reviewer updated their rating to weak accept. Therefore, the AC recommends a weak accept for this paper.